# Expression Profiling of *Pdx1*, *Ngn3*, and *MafA* in the Liver and Pancreas of Recovering Streptozotocin-Induced Diabetic Rats

**DOI:** 10.3390/genes13091625

**Published:** 2022-09-10

**Authors:** Amani M. Al-Adsani, Anoud N. Al-Otaibi, Sahar A. Barhoush, Khaled K. Al-Qattan, Suzanne A. Al-Bustan

**Affiliations:** 1Department of Biological Sciences, Faculty of Science, Kuwait University, Kuwait City 13060, Kuwait; 2Sheikh Abdullah Al-Salem Cultural Centre, Salmiya 22065, Kuwait

**Keywords:** diabetes, pancreas development, β-cell regeneration, transcription factors, transdifferentiation

## Abstract

Studies in animal diabetic models have demonstrated the possibility of islet regeneration through treatment with natural extracts, such as *Allium sativum* (garlic). This study aimed to investigate the effect of garlic extract (GE) on the expression of three genes (*Ngn3*, *Pdx1*, and *MafA)* in the pancreas and liver of diabetic rats. Thirty-two rats were divided into two groups, streptozotocin (STZ)-induced diabetic rats (*n* = 16) and healthy rats (*n* = 16). Both groups were subdivided into GE-treated (*n* = 8), and those administered 0.9% normal saline (NS) (*n* = 8) for 1 week (*n* = 4) and 8 weeks (*n* = 4). In the pancreas of diabetic rats treated with GE for 1 week, all three genes, *Ngn3*, *Pdx1*, and *MafA*, were significantly upregulated (*p* ≤ 0.01, *p* ≤ 0.05, and *p* ≤ 0.001, respectively) when compared to diabetic rats treated with NS only. However, after eight weeks of GE treatment, the expression of all three genes decreased as blood insulin increased. In the liver, only *Pdx1* expression significantly (*p* ≤ 0.05) increased after 8 weeks. The significant expression of *Ngn3*, *Pdx1*, and *MafA* in the pancreas by week 1 may have induced the maturation of juvenile β-cells, which escaped the effects of STZ and caused an increase in serum insulin.

## 1. Introduction

Diabetes mellitus (DM) affects many people worldwide, and its prevalence is on the rise, with an expected increase from 2.8% in 2000 to 4.4% in 2030 [1]. In the Middle East, specifically in Kuwait, approximately 19.1% of the population suffers from this disease, and 13.5% are currently prediabetics [2]. Patients with Type-1 diabetes (T1D) resulting from autoimmune destruction of β-cells exhibit hyperglycaemia due to low levels of blood insulin. The available treatments typically include exogenous insulin sources. However, alternative natural products such as garlic have been proven to be effective in alleviating the symptoms of diabetes and restoring normoglycaemia [3]. Exploring the mechanism by which insulin production increases may elucidate the differential gene expression involved in β-cell neogenesis.

In the past two decades, the development of pancreatic endocrine cells, specifically β-cells, has been extensively investigated both in vitro and in vivo using various animal models [4,5]. Consequently, key transcription factors involved in β-cell differentiation have been identified including neurogenin-3 (*Ngn3*), pancreatic and duodenal homeobox 1 (*Pdx1*), and V-maf musculoaponeurotic fibrosarcoma oncogene homolog A (*MafA*) (referred to as NPM factors) [6]. At the top of the committed pancreatic endocrine cell lineage is the basic helix-loop-helix (bHLH) transcription factor *Ngn3*, which causes cells to differentiate into all endocrine cell lineages [7]. Therefore, *Ngn3*-positive cells are considered endocrine progenitor cells that will consequently differentiate into islet cells upon the differential expression of other downstream genes. In the developing mouse pancreas, *Ngn3* expression peaks around E15.5, then greatly decreases and almost diminishes in the adult pancreas [8]. At E13.5, the basic leucine zipper transcription factor *Ngn3* is expressed in the second transition stage of the developing pancreas in insulin-producing cells. In mature islets, *Ngn3* is restricted to β-cells and is an important regulator of insulin expression [9,10]. *Pdx1* is a homeodomain transcription factor that is expressed at different time points: first, during early development of the mouse pancreas around E8.5, then in multipotent cells during endocrine lineage specification, and later in immature and mature β-cells at E18.5, as it maintains their proliferation, function, and survival [11]. *Pdx1* is also responsible for the regulation of insulin, glucose transporter 2 (*Glut2)*, and NK6 homeobox 1 (*Nkx6.1*). When expressed in β-cells, it binds to the A-box, a highly conserved element in the upstream region of the insulin promoter [12].

The role of these three NPM factors is not only limited to pancreatic organogenesis, where they induce β-cell development and maturation, but also in the direct reprogramming of insulin-producing cells from non-β-cells [13]. Zhou et al. [14] conducted a study in which they showed the combined ability of *Ngn3*, *MafA*, and *Pdx1* to induce β-like cells from pancreatic exocrine cells of adult mice with the ability to ameliorate hyperglycaemia through insulin secretion. Adding a fourth transcription factor, paired box 4 (*Pax4*), to the NPM genes and growth factors in human pancreatic exocrine cells resulted in the generation of functional β-like cells that were able to reduce blood glucose levels after transplantation in adult diabetic mice [15]. Transgenic mice conditionally expressing *Ngn3* and *Pdx1* induced the convergence of *Ngn3*-positive progenitor cells into insulin-positive cells and transdifferentiation of glucagon-positive α-cells into β-cells [16]. Forced expression of all three NPM factors in acinar cells resulted in their convergence to β-cells [17]. NPM-mediated transdifferentiation from fully differentiated nonpancreatic adult cell types has also been demonstrated in rat liver [18] and intestinal crypts [19].

Our laboratory established a recovering streptozotocin (STZ)-induced diabetic rat model after garlic extract (GE) (*Allium sativum*) treatment. These rats were able to attenuate the symptoms of diabetes by increasing insulin production [3]. This study utilised an established STZ-induced diabetic rat model and investigated the expression of the three transcription factors *Ngn3*, *Pdx1*, and *MafA* using real-time PCR in the pancreas and liver of recovering diabetic rats. The pancreas could remain a source of insulin production despite STZ treatment specifically targeting β-cells to annihilate them [20]. Furthermore, since both the liver and pancreas arise from the foregut endoderm [8,21], the liver is a possible target of investigation where insulin may be produced. *Ngn3* is a suitable marker for potential β-cell differentiation because it is a proendocrine factor expressed in endocrine progenitor cells [22]. *Pdx1* is also involved in the early development of the pancreas and regulates insulin expression in mature β-cells [12]; therefore, it serves as an excellent marker for both differentiated functional β-cells and differentiation events that lead to new β-cells or insulin-producing cell formation, possibly in both the pancreas and liver. *MafA*, on the other hand, may serve as a marker for already differentiated functional β-cells or insulin-producing cells because it is expressed only in differentiated β-cells [23]. We hypothesised that garlic may induce the differential expression of β-cell differentiation genes in STZ-DM rats treated with GE, leading to increased insulin production as a result of β-cell development and/or insulin gene regulation in the pancreas and/or liver.

## 2. Materials and Methods

### 2.1. Animal Model

A total of 38 male Sprague Dawley rats (Harlan Laboratories, Derby, UK) with an initial bodyweight of 120–150 g were used in this study. The animals were handled according to the instructions for the care and use of laboratory animals. The rats were housed in the Animal Care and Breeding Unit at the Department of Biological Sciences, Kuwait University, under moderate climatic conditions (22–24 °C ambient temperature, 30–35% ambient humidity, and 12:12 h natural light/dark cycle) and provided with a standard rodent diet (Special Diet Services, Witham, UK) and tap water ad libitum.

### 2.2. Induction of Diabetes and Treatment with GE

T1D was induced in 16 rats by intraperitoneal injection of STZ (60 mg/kg) in 0.5 mL of citrate buffer after an overnight fast. The remaining 16 nondiabetic rats were injected with 0.5 mL of citrate buffer only. To ascertain the induction of T1D (or normality), fasting blood glucose (FBG) levels of all rats (*n* = 32) were measured using a portable glucometer (One Touch UltraEasy—LifeScan, Buckinghamshire, UK) from 0.1 mL of tail blood after an overnight fast at day 6 postinjection with either STZ or citrate buffer. STZ-injected rats with FBG levels higher than 16.5 mmol/L were considered diabetic (*n* = 16), while citrate buffer-injected rats with FBG levels < 6 mmol/L (*n* = 16) were considered normal. All 32 healthy and diabetic rats were administered daily treatments of 0.5 mL of 0.9% saline or GE (500 mg/0.5 mL/kg b.wt) as follows: eight normal healthy rats in group A were treated with saline (NR-NS), eight healthy rats in group B were treated with GE (NR-GE), eight diabetic rats in group C were treated with saline (DR-NS), and eight diabetic rats in group D were treated with GE (DR-GE). Three untreated healthy rats and three untreated diabetic rats were used as controls for baseline gene expression profiling of the target genes.

### 2.3. Tissue Sample Collection and RNA Extraction

Following the induction of anaesthesia with a mixture of ketamine (9 mL, 10%, Dutch farm Nedar, Host den Berg, Holland) and xylazine (1 mL, 10%, Interchemie, Venary, Holland), four rats from each group (A, B, C, and D) were sacrificed after 1 week (W1) and 8 weeks (W8) for analysis at two different time points. Three untreated control and three untreated diabetic rats were sacrificed on day 3 following diabetes induction for baseline measurements. Under aseptic conditions and through an abdominal midline incision, blood was collected via a cardiac puncture. Blood serum was then isolated and stored at −20 °C for later analysis. Next, the duodenal portion of the pancreas and part of the liver were exposed, excised, weighed, and rinsed with diethyl pyrocarbonate (DEPC) water, and 0.4 g of tissue sample was used for RNA extraction. To ensure high quality and integrity of pancreatic RNA, extraction was performed immediately. However, liver tissue was stored in liquid nitrogen at −80 °C for later RNA extraction. The tissue samples were homogenised using 1 mL TRIzol^®^ Reagent (Invitrogen, CA, USA) per 50–100 mg of tissue, and total RNA was extracted from both the pancreatic and liver samples using the TRIzol^®^ method according to the manufacturer’s instructions and stored at −80 °C. Each sample was analysed for yield, concentration, and purity by measuring the A260/A280 absorbance ratio using a Nanodrop 8000 spectrophotometre (Thermo Fisher Scientific, Waltham, MA, USA). RNA quality was determined using an Agilent RNA 6000 Nano Kit and an Agilent 2100 Bioanalyzer (Agilent Technologies, Santa Clara, CA, USA).

### 2.4. Measurement of Blood Glucose and Insulin Concentrations after Treatment with GE or Normal Saline

FBG levels of all rats (*n* = 32) were measured from 0.1 mL of tail blood after an overnight fast at the end of weeks 1 and 8 of treatment in the normal control (*n* = 16) and STZ-induced diabetic rats (*n* = 16). Insulin levels were also determined by enzyme immunoassay using an ELISA kit following the manufacturer’s instructions (SPI bio, Bertin Pharma, Montigny-le-Bretonneux, France).

### 2.5. Conversion of RNA to cDNA

cDNA was prepared from RNA samples (*n* = 16) extracted from 8 DR-NS and 8 DR-GE by treating the samples with DNase I for 15 min followed by high-capacity RNA to cDNA synthesis kit (Applied Biosystems, Carlsbad, CA, USA). RNA was diluted to a concentration of 1 µg of RNA per 20 μL of reaction that included 10 µL of 2X RT buffer mix containing dNTPs, random octamers and oligo dT-16, 1 µL of 20X RT enzyme mix, and 9 μL of RNA. All samples were incubated at 37 °C for 60 min, then at 95 °C for 5 min, ending with 4 °C using an automated thermocycler (Applied Biosystems Fast Thermal Cycler Version 1.01, Life Technologies, Carlsbad, CA, USA), then stored at −20 °C.

### 2.6. Real-Time PCR

Thirty-two cDNA samples (sixteen from DR-NS and sixteen from DR-GE) were used to analyse *Pdx1*, *Ngn3*, and *MafA* expression levels with the housekeeping gene *β-actin* (*β-actin*) as an internal control. Real-time PCR of all cDNA was carried out according to the manufacturer’s protocol in a final volume of 25 μL using predesigned expression TaqMan^®^ assays (Applied Biosystems, Pleasanton, CA, USA) with labelled probes and primers for each gene: *β-actin*: Rn00667869_m1 (VIC), *Ngn3*: Rn00572583_s1 (FAM), *Pdx1-*: Rn00755591_m1 (FAM), and *MafA*: Rn00845206_s1 (FAM), using an ABI 7900HT real-time instrument. Gene expression levels were expressed as relative mRNA levels compared to internal control levels after normalisation to *β-actin*, based on ΔΔCT. Triplicates of each sample were used and subjected to statistical analysis to confirm the increase or decrease in gene expression.

### 2.7. Statistical Analysis

FBG, serum insulin, and pancreas and liver real-time PCR analyses are presented as mean ± SE in bar graphs. Statistical comparisons were carried out using two-way analysis of variance (ANOVA) followed by Fisher’s LSD test for biophysical and biochemical analyses using GraphPad Prism version 9.2.0.332. Differences among the analyses were considered statistically significant at *p* < 0.05.

Statistical analysis of the real-time PCR data was performed using the comparative CT method (2^−**ΔΔCt**^) to represent relative gene expression levels. Cycle threshold (CT) values were used as a reference to compare the gene expression profiles in the pancreatic and liver samples from diabetic rats treated with saline (DR-NS) or GE (DR-GE) over the eight-week period.

## 3. Results

### 3.1. FBG and Serum Insulin Concentrations

The effects of GE on the biological and biochemical parameters of STZ-induced diabetic rats were consistent with those reported previously [24]. GE treatment significantly increased the bodyweight of DR, reflecting their gradual recovery, in contrast to DR treated with only NS, which showed a significant decline in bodyweight by week 8 (Figure 1). Moreover, 8 weeks after starting the treatment, DR-GE displayed significantly lower blood glucose levels (*p* ≤ 0.001) than DR-NS (Figure 2). Insulin levels increased significantly (*p* ≤ 0.001) in DG after 8 weeks of GE treatment compared to week 1 and in DR-NS rats (*p* ≤ 0.001) at week 8 (Figure 3).

### 3.2. Real Time-PCR Profiling

#### 3.2.1. Pancreas Gene Expression Profiling

The gene profiles of *Pdx1*, *MafA*, and *Ngn3* in the pancreas of both DR-NS and DR-GE rats were measured after treatment at weeks 1 and 8 (Figure 4). After only one week of GE treatment, DR-GE *Ngn3* showed significantly higher expression (*p* ≤ 0.01) (Figure 4A) with a 6.81-fold change compared to DR-NS. However, as the treatment progressed to 8 weeks, *Ngn3* expression decreased in the DR-GE group, reaching levels similar to those in the DR-NS group, without any significant difference (*p* ≥ 0.05). The expression of *Pdx1* was only significantly increased (*p* ≤ 0.05) (1.9-fold change) in DR after 1 week of GE treatment (DR-GE) (Figure 4B). After 8 weeks of GE treatment, *Pdx1* expression significantly decreased (*p* ≤ 0.05). DR treated with normal saline (DR-NS) did not display any significant variation in *Pdx1* expression between weeks 1 and 8. GE treatment for only one week induced a significant increase (*p* ≤ 0.001) in *MafA* expression in diabetic rats (DR-GE) (Figure 4C). Conversely, no significant differences were detected between the same time points in the DR-NS group. After treatment with GE for 8 weeks (DR-GE), *MafA* expression significantly decreased (*p* ≤ 0.001) from 52.3-fold at week 1 to 3.19-fold at week 8.

#### 3.2.2. Liver Gene Expression Profiling

The expression levels of the NPM transcription factors were also investigated in the livers of both DR-NS and DR-GE rats (Figure 5). GE did not result in any differences in the expression levels of either *Ngn3* (Figure 5A) or *MafA* (Figure 5C) after 1 and 8 weeks of GE treatment. *Pdx1*, on the other hand, was significantly more upregulated (*p* ≤ 0.05) (99.31-fold change) in DR-GE compared to DR-NS (19.92-fold change) (Figure 5B). In addition, after eight weeks of GE treatment, *Pdx1* expression significantly decreased (*p* ≤ 0.05) from a 99.31-fold change to a 11.01-fold change in DR-GE. No significant difference was detected in the DR-NS group after eight weeks of NS administration.

## 4. Discussion

Natural remedies have been used to treat many ailments for several years. Herbal medicines have been shown to improve β-cell regeneration and function [25], and garlic has been demonstrated to reduce blood glucose in diabetic rats [24]. The source of insulin in these recovering diabetic rats may originate from a few organs, mainly the pancreas and the liver [26]. However, the findings in this study support the idea that insulin-producing cells may have originated from the pancreas of DR patients in response to treatment with GE.

In the liver, GE treatment did not significantly affect the expression levels of *Ngn3* and *MafA*. However, *Pdx1* expression increased significantly (*p* ≤ 0.05) in DR treated with GE for 1 week compared to the control DR group that was administered NS only (Figure 5B). Ferber et al. [27] were the first to demonstrate the possibility of liver-to-pancreas cell conversion by ectopic *Pdx1* expression in STZ-induced diabetic mice, where an increase in biologically active insulin was able to ameliorate hyperglycaemia. Adenovirus-mediated *Pdx1* gene therapy was also shown to reverse the symptoms of T1D in cyclophosphamide-accelerated diabetes in nonobese diabetic (CAD-NOD) mice [28]. Further studies were conducted to improve the reprogramming of some liver cells to β-cells using *Pdx1* in addition to other transcription factors, including Neuronal differentiation factor (*NeuroD*), *MafA*, *Ngn3*, *Nkx6.1*, and *Pax4* [29]. The possibility of GE treatment reprograming liver cells to form insulin-producing cells cannot be excluded. Further investigation is needed to determine the presence of insulin in this tissue.

In the pancreas, after only 1 week of treatment with GE, the expression of *Ngn3*, *Pdx1*, and *MafA* in the DR-GE group was significantly increased (*p* ≤ 0.05) compared to that in the DR-NS group (Figure 4). However, after eight weeks, the expression levels of all three genes decreased. The initial increase in the expression of all three NPM transcription factors in the DR in response to GE may have been the driving force triggering the formation or maturation of insulin-producing cells, consequently inducing insulin gene expression in the pancreas. Although all three genes showed a significant increase after one week, their expression levels varied.

Variations in the expression of all three NPM transcription factors in pancreatic tissues suggest that GE may act on different cell types, mainly progenitor or immature β-cells. GE may have induced the differentiation of putative progenitor cells to give rise to mature β-cells in the injured pancreas. Moreover, since mature β-cells were destroyed by STZ through the expression of Glut-2, some juvenile β-cells may have escaped the effects of STZ and, in turn, were induced to mature after treatment with GE. Mature β-cell identity was restored by *MafA*, which showed the highest expression (*p* ≤ 0.001) because of significantly elevated (*p* ≤ 0.05) *Pdx1* levels. *Ngn3* was the factor with the second most significantly increased expression (*p* ≤ 0.01). This transcription factor is considered the major regulator of the formation and differentiation of endocrine progenitor precursors [30]. Bechard et al. [31] suggested that the low level of *Ngn3* expression in mitotic endocrine progenitors initiates endocrinogenesis. Finally, *Pdx1* was the third most significantly expressed gene (*p* ≤ 0.05). *Pdx1 could* have played a role in the endocrine lineage specification of progenitor cells. Moreover, *Pdx1* may have induced β-cell maturation [13] and insulin gene upregulation by binding to the upstream region of the insulin gene promoter [12]. Insulin gene expression is synergistically activated by the coordinated binding of *Pdx1* and other transcription factors, including *MafA* and *NeuroD1*, to the insulin promoter when blood glucose levels are elevated [32]. In a recent study [33], it was demonstrated that low levels of *Pdx1* and *MafA* are needed for islet cell function and insulin release where maintained high levels of both genes lead to dysregulation in the metabolic processes. This may also explain the downregulation of *Pdx1* and *MafA* by week 8. It seems like one week of GE treatment was probably enough to cause an initial significant increase in all three transcription factors (*Ngn3*, *Pdx1*, and *MafA*) and trigger the necessary downstream molecular effects that ultimately resulted in a significant (*p* ≤ 0.001) increase in blood insulin levels after the continuation of GE treatment for 8 weeks (Figure 3), where all three genes were consequently downregulated. A proposed model of the effect of differential expression of the three NPM genes on pancreatic cells of DR treated with GE is illustrated in Figure 6.

## 5. Conclusions

Treatment of STZ-induced diabetic rats with GE for 8 weeks had a significant effect on the expression profiles of the three selected NPM genes in the pancreas and *Pdx1* in the liver. After 1 week of treatment, GE induced a significant increase in the expression levels of the three key genes, *Ngn3*, *Pdx1*, *and MafA*, which control β-cell development, suggesting that the formation of insulin-producing cells was initiated in the pancreas. After continuation of GE treatment for eight weeks, all NPM transcription factors decreased as blood insulin levels increased. These newly formed β-like cells may have originated from several sources in the injured pancreas. The high expression of *Ngn3*, *Pdx1*, and *MafA* by week 1 may have acted as a driving force that pushed pancreatic progenitor cells into endocrine lineages or induced the maturation of juvenile β-cells. Further investigation is needed to localise the source of insulin in the pancreas of recovering STZ-induced DR in response to GE through immunohistochemistry using various pancreatic endocrine and β-cell markers.

## Figures and Tables

**Figure 1 genes-13-01625-f001:**
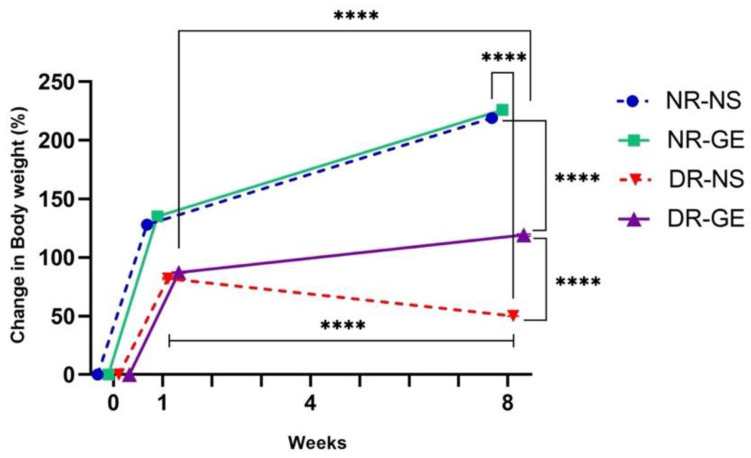
Percentage changes in bodyweight levels for all rat groups after 1 and 8 weeks of GE treatment. ****: *p* ≤ 0.001.

**Figure 2 genes-13-01625-f002:**
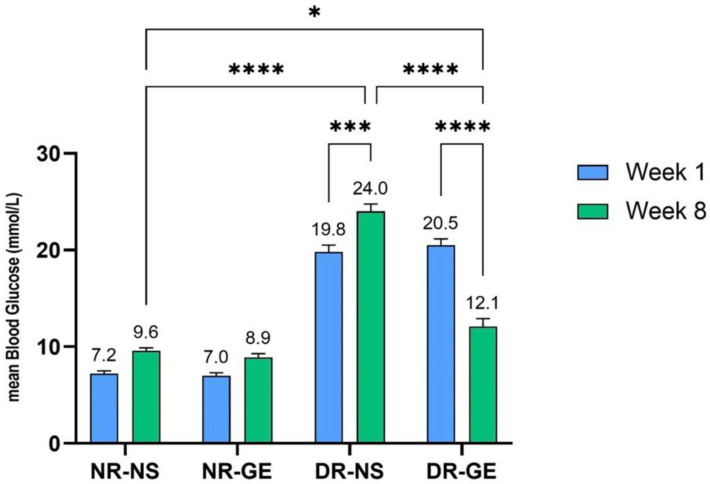
Changes in fasting blood glucose (mmol/L) for all rat groups after 1 and 8 weeks of GE treatment. *: *p* ≤ 0.05, ***: *p* ≤ 0.01, ****: *p* ≤ 0.001.

**Figure 3 genes-13-01625-f003:**
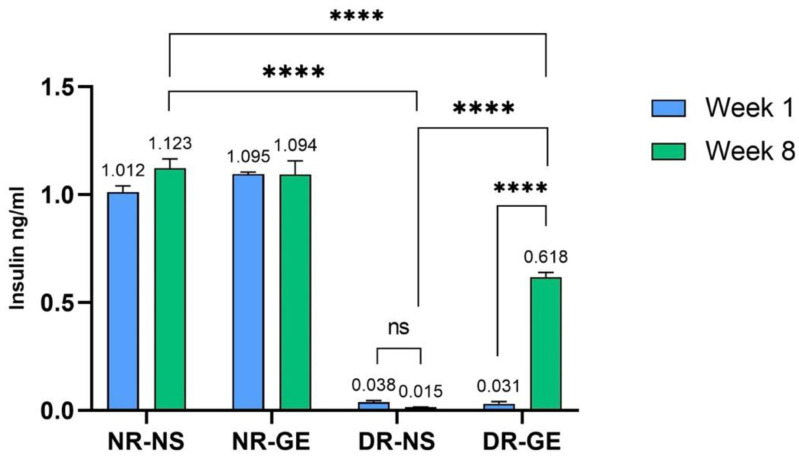
Changes in serum insulin levels (ng/mL) for all rat groups after 1 and 8 weeks of GE treatment. ns: no significance, ****: *p* ≤ 0.001.

**Figure 4 genes-13-01625-f004:**
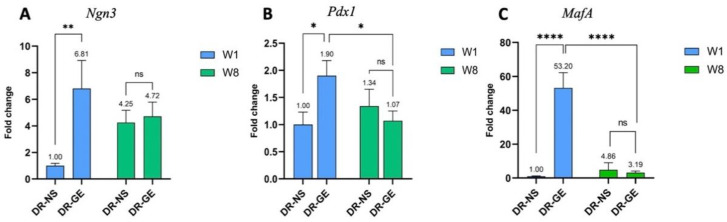
Expression profiles of the three NPM factors in the pancreas of diabetic rats treated with normal saline (NS) or garlic extract (GE) for up to 8 weeks. All three genes (**A**) *Ngn3*, (**B**) *Pdx1*, and (**C**) *MafA* showed significant variable expression levels in diabetic rats following GE treatment. ns: no significance, *: *p* ≤ 0.05, **: *p* ≤ 0.01, ****: *p* ≤ 0.001.

**Figure 5 genes-13-01625-f005:**
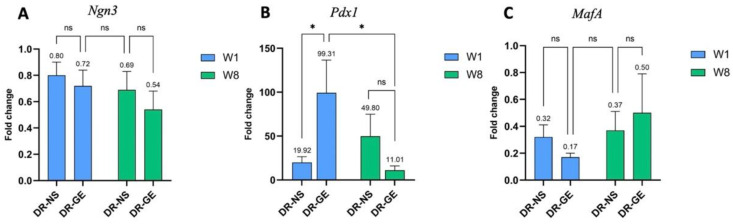
NPM factor expression in the liver of diabetic rats (DR) treated with either normal saline (NS) or garlic extract (GE) for 1 and 8 weeks. (**A**) *Ngn3* and (**C**) *MafA* did not show any significant difference in expression levels. (**B**) *Pdx1* was significantly expressed after 1 week of GE treatment compared to that in DR given only NS. ns: no significance, *: *p* ≤ 0.05.

**Figure 6 genes-13-01625-f006:**
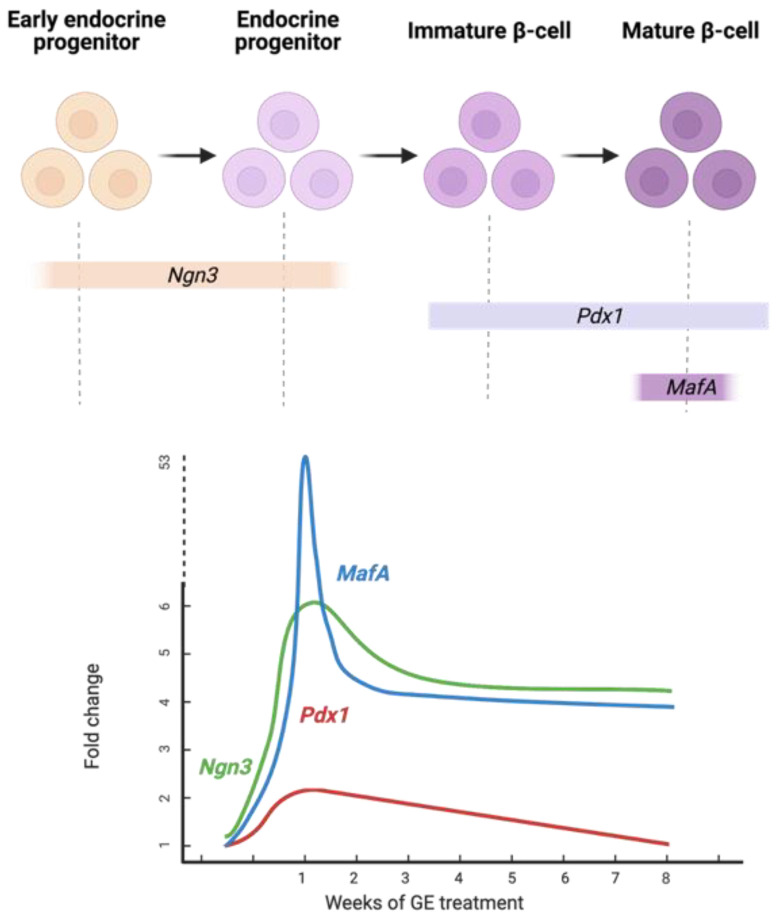
Differential expression profile of NPM genes in the recovering pancreas of STZ-induced diabetic rats after 1 and 8 weeks of GE treatment. *MafA* expression is the highest at week 1, followed by *Pdx1* and then *Ngn3*. Insulin-producing cells may have originated from the differentiation of immature β-cells that escaped the effects of STZ and possibly were induced by both *MafA* and *Pdx1* to give rise to mature β-cells. Created with BioRender.com, accessed on 14 June 2022. GE, garlic extract.

## Data Availability

The authors declare that all the data supporting the findings of this study are available within the article or from the corresponding author upon reasonable request.

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
