# Peer review of "Expression Profiling of Pdx1, Ngn3, and MafA in the Liver and Pancreas of Recovering Streptozotocin-Induced Diabetic Rats"

_genes, 2022, doi:10.3390/genes13091625_

Round 1
Reviewer 1 Report
The manuscript is straightforward, well written and although very brief, it introduces very clearly the role of the three transcription factors and the results obtained. However, there are some concerns. After introducing in detail that the expression of the three genes could have a different effect in different pancreas cell types, authors simply tested the expression of Pdx1, MafA and Ngn3 in the whole pancreas and liver; this type of investigation could be just the average of expression of the 3 genes and has clear limitations that should be clearly discussed. In addition, GE plays a key role in the manuscript, but its characterization is not reported: how was it prepared? How was it tested? This is absolutely relevant for the impact of the manuscript.
Major points
Authors wrote “This study utilised an established STZ-induced diabetic rat model and investigated the expression of the three transcription factors Ngn3, Pdx1, and MafA in the pancreas and liver of recovering diabetic rats.” In this sentence, it is unclear what part of the pancreas was investigated. Similarly, in the Results section (lines 196/197) it is unclear what part of the pancreas was investigated and what is the methodological approach used. This should be immediately clear.
Authors wrote “We hypothesised that garlic may induce the differential expression of β-cell differentiation genes in STZ-DM rats 85 treated with GE, leading to increased insulin production as a result of β-cell development and/or insulin gene regulation in the pancreas and/or liver.” Why authors choose GE? What active molecules could account for its action? Was the effect of GE previously described in the literature? These aspects should be present in the Introduction
In the Introduction, here is no mention of the methodology used to assay NMP factors
Minor points
The sentence “However, alternative natural products such as garlic have been proven to be effective in elevating the symptoms of diabetes” seems an incoherent sentence, perhaps the authors would mean “…have been proven to be effective in alleviating the symptoms of diabetes…”
Author Response
Dear referee,
Thank you for reviewing the manuscript and for your valuable feedback. Below is point by point response to the raised questions and comments.
Best regards,
Dr. Amani Al-Adsani
Major points
1. Authors wrote “This study utilised an established STZ-induced diabetic rat model and investigated the expression of the three transcription factors Ngn3, Pdx1, and MafA in the pancreas and liver of recovering diabetic rats.” In this sentence, it is unclear what part of the pancreas was investigated. Similarly, in the Results section (lines 196/197) it is unclear what part of the pancreas was investigated and what is the methodological approach used. This should be immediately clear.
- The duodenal lobe of the pancreas was excised. This is because we wanted to be consistent in sampling and this part of the pancreas is easily identifiable in both diabetic and healthy rats. This is now clarified in the manuscript line 120. As for the methodological approach for RNA extraction, it is also clarified in the methodology section line 124-127.
2. Authors wrote “We hypothesised that garlic may induce the differential expression of β-cell differentiation genes in STZ-DM rats treated with GE, leading to increased insulin production as a result of β-cell development and/or insulin gene regulation in the pancreas and/or liver.” Why authors choose GE? What active molecules could account for its action? Was the effect of GE previously described in the literature? These aspects should be present in the Introduction
- Garlic extract (GE) preparation: GE was prepared from a single batch of garlic (Allium sativum L.) cloves that were purchased from the local market as previously described (Ali and Mohammad et al., 1985). About 150 g of fresh garlic was peeled and homogenized using a blender in 100 ml of cold sterile 0.9% NaCl. The mixture was filtered through cheesecloth then centrifuged at 15000 rpm for 15 min. The clear supernatant was made up to 300 ml by adding ice-cold normal saline and stored in aliquots of a single daily servings at -20°C for future use. Each day of treatment the required volume of the garlic extract was thawed to room temperature and administered as planned.
- The antidiabetic effect of GE was previously tested by many groups. Previous publications from our lab describing GE preparation and the hypoglycemic effects of GE are:
1. Thomson, M., Al-Qattan, K. K., Js, D., & Ali, M. (2016). Anti-diabetic and anti-oxidant potential of aged garlic extract (AGE) in streptozotocin-induced diabetic rats. BMC complementary and alternative medicine, 16, 17. https://doi.org/10.1186/s12906-016-0992-5
2. Drobiova, H., Thomson, M., Al-Qattan, K., Peltonen-Shalaby, R., Al-Amin, Z., & Ali, M. (2011). Garlic increases antioxidant levels in diabetic and hypertensive rats determined by a modified peroxidase method. Evidence-based complementary and alternative medicine: eCAM, 2011, 703049. https://doi.org/10.1093/ecam/nep011
3. Thomson, M., Al-Qattan, K. K., Bordia, T., & Ali, M. (2006). Including garlic in the diet may help lower blood glucose, cholesterol, and triglycerides. The Journal of nutrition, 136(3 Suppl), 800S–802S. https://doi.org/10.1093/jn/136.3.800S
Since the antidiabetic effects of GE was well established in our lab, we seeked to identify the differential expression of genes that are involved in beta cell development in the pancreas. The liver was also chosen because many studies have suggested its ability to give rise to insulin-producing cells because both the liver and pancreas arise from the foregut endoderm during development.
As was mentioned in the introduction section pages 69-73: “Our laboratory established a recovering streptozotocin (STZ)-induced diabetic rat model after garlic extract (GE) (Allium sativum) treatment. These rats were able to attenuate the symptoms of diabetes by increasing insulin production [3]. This study utilised an established STZ-induced diabetic rat model and investigated the expression of the three transcription factors Ngn3, Pdx1, and MafA in the pancreas and liver of recovering diabetic rats”.
Consequently, the focus in this manuscript is on the recovering diabetic rats in response to GE and not the GE itself. Since the effects of GE on STZ-induced diabetic rats are well established by our group, we simply utilized the recovering diabetic rats in response to GE as a model to focus on beta cell neogenesis in the recovering diabetic rats.
3. In the Introduction, here is no mention of the methodology used to assay NMP factors
- The methodology to assay NMP factors was clarified in the introduction in line 73.
Reviewer 2 Report
Comments:
1. Page 3, lines 140-141: “RT-PCR using hexamer primers and cDNA synthesis kits in a total volume of 140 20 μL. The reaction mixture was as follows: 2 μL of random primers (100 ng/μL)”. Do authors use different primers or writing faults?
2. Page 4, line 153: “Sixteen cDNA samples (16 from DR-NS and 16 from DR-GE)” should change to Thirty-two cDNA samples?
3. Page 4, line 158: “PDX-1: Rn00755591_m1 (FAM)” should change to Pdx-1: Rn00755591_m1 (FAM)”.
4. Please provide the individual body weight (g) in Figure 1.
5. To confirm the effect of garlic extract, authors should perform a function experiment, such as an Intraperitoneal glucose tolerance test (IPGTT) or Intraperitoneal Insulin Tolerance Test (IPITT). Also, the beta-cell mass should be measured after the rats were sacrificed.
6. In Figure 4B, “The expression of Pdx1 was only significantly increased (P ≤ 0.05) (1.9-fold change) in DR after 1 week of GE treatment (DR-GE) (Figure 4B). After 8 weeks of GE treatment, Pdx1 expression significantly decreased (P ≤ 0.05)”. From Figure3, the plasma insulin level significantly increased after 8 weeks of GE treatment, suggesting that there are more beta cells inside. As we know mature beta-cell also express Pdx1, but, after 8 weeks of GE treatment, Pdx1 expression significantly decreased. How to explain the initial possible mechanism?
7. In discussion, line 241, if they want to address “The possibility of GE treatment reprogramming liver cells to form insulin-producing cells”. They should do beta cell markers (insulin/Pdx1) immunofluorescence staining on the liver of NR-GE groups.
8. To figure out the source of those new beta cells, they should measure the gene expression levels of Nkx6.1 (mature beta cells marker) after 1 and 8 weeks of GE treatment.
9. In Figure 4, they use whole pancreas tissue as samples for gene expression measurement, it’s hard to exclude the other cell type (exocrine cells) interference due to pancreas heterogeneity. They should isolate islet cells for the samples.
10. To address their hypothesis, there are many experiments they can do, such as primary islet cell and hepatocytes in vitro cultured with GE then collect the sample for RT-PCR, measurement of those transcription factors (Ngn3, Pdx1, MafA, and Nkx6.1).
11. Please showed all the individual dots in the bar graph.
Author Response
Dear referee,
Thank you for reviewing the manuscript and for your valuable feedback. Below is point by point response to the raised questions and comments.
Best regards,
Dr. Amani Al-Adsani
Response for referee 2:
1. Page 3, lines 140-141: “RT-PCR using hexamer primers and cDNA synthesis kits in a total volume of 20 μL. The reaction mixture was as follows: 2 μL of random primers (100 ng/μL)”. Do authors use different primers or writing faults?
- Random octamers that were included in the cDNA synthesis kit were used. The paragraph was accordingly corrected and clarified in the text from line 140.
2. Page 4, line 153: “Sixteen cDNA samples (16 from DR-NS and 16 from DR-GE)” should change to Thirty-two cDNA samples?
- Sixteen was replaced with thirty two.
3. Page 4, line 158: “PDX-1: Rn00755591_m1 (FAM)” should change to Pdx-1: Rn00755591_m1 (FAM)”.
- PDX-1: Rn00755591_m1 (FAM) was replaced with Pdx1: Rn00755591_m1 (FAM).
4. Please provide the individual body weight (g) in Figure 1.
The table provided shows body weight percentages with the standard error of the mean (SEM), values in (g) are available but unfortunately the person is on holiday and does not have access to the data).
- Rats body weight were measured and saved as percentages with SEM.
BODY WEIGHT (GAIN/LOSS %)
Group Pre STZ Post STZ (n=12 ) Week 1 (n= 12) Week 8 (n=4)
DR-NS 100 89 ± 0.879 82 ± 0.743 50 ± 0.087
DR-GE 100 83.19 ± 0.789 87.2 ± 0.398 119.21 ± 0.934
(DR-NS diabetic rats treated with saline, DR-GE diabetic rats treated with garlic extract).
5. To confirm the effect of garlic extract, authors should perform a function experiment, such as an Intraperitoneal glucose tolerance test (IPGTT) or Intraperitoneal Insulin Tolerance Test (IPITT). Also, the beta-cell mass should be measured after the rats were sacrificed.
- The antidiabetic effect of GE was previously tested by many groups. Publications from our lab describing the hypoglycemic effects of GE include:
1. Thomson, M., Al-Qattan, K. K., Js, D., & Ali, M. (2016). Anti-diabetic and anti-oxidant potential of aged garlic extract (AGE) in streptozotocin-induced diabetic rats. BMC complementary and alternative medicine, 16, 17. https://doi.org/10.1186/s12906-016-0992-5
2. Drobiova, H., Thomson, M., Al-Qattan, K., Peltonen-Shalaby, R., Al-Amin, Z., & Ali, M. (2011). Garlic increases antioxidant levels in diabetic and hypertensive rats determined by a modified peroxidase method. Evidence-based complementary and alternative medicine: eCAM, 2011, 703049. https://doi.org/10.1093/ecam/nep011
3. Thomson, M., Al-Qattan, K. K., Bordia, T., & Ali, M. (2006). Including garlic in the diet may help lower blood glucose, cholesterol, and triglycerides. The Journal of nutrition, 136(3 Suppl), 800S–802S. https://doi.org/10.1093/jn/136.3.800S
- In this study beta cell mass was not measured since the aim of the study was to only investigate NMP factors using real-time PCR in the pancreas and liver. The mass was measured in another following completed study (from another grant) where a description and analysis of the regenerating islets were included in a different submitted manuscript.
6. In Figure 4B, “The expression of Pdx1 was only significantly increased (P ≤ 0.05) (1.9-fold change) in DR after 1 week of GE treatment (DR-GE) (Figure 4B). After 8 weeks of GE treatment, Pdx1 expression significantly decreased (P ≤ 0.05)”. From Figure 3, the plasma insulin level significantly increased after 8 weeks of GE treatment, suggesting that there are more beta cells inside. As we know mature beta-cell also express Pdx1, but, after 8 weeks of GE treatment, Pdx1 expression significantly decreased. How to explain the initial possible mechanism?
- The up regulation of insulin gene by week 1 and the abundance of insulin protein stored in β cells could have led to the down regulation of insulin transcription factors like Pdx1 and MafA. This may explain their lowered expression by week 8 in the pancreas. The difference between the half-life of mRNA and protein could also be a factor because the average mRNA half-life is 7.1 hour for genes related to metabolism whereas the average protein half-life is 1-2 days, and this might be more prominent by week 8 as insulin concentration becomes significantly higher (P ≤ 0.001) in blood serum. Furthermore, a recent study (Nasteska et al., 2021) showed that low levels of Pdx1 and MafA is needed for islet cell function and insulin release since maintained high levels of both genes lead to dysregulation in the metabolic processes due to dysregulation of their gene pathways.
(Nasteska, D., Fine, N., Ashford, F. B., Cuozzo, F., Viloria, K., Smith, G., Dahir, A., Dawson, P., Lai, Y. C., Bastidas-Ponce, A., Bakhti, M., Rutter, G. A., Fiancette, R., Nano, R., Piemonti, L., Lickert, H., Zhou, Q., Akerman, I., & Hodson, D. J. (2021). PDX1LOW MAFALOW β-cells contribute to islet function and insulin release. Nature communications, 12(1), 674. https://doi.org/10.1038/s41467-020-20632-z)
7. In discussion, line 241, if they want to address “The possibility of GE treatment reprogramming liver cells to form insulin-producing cells”. They should do beta cell markers (insulin/Pdx1) immunofluorescence staining on the liver of NR-GE groups.
- In the conclusion of the current study line 304, we stated that “Further investigation is needed to localise the source of insulin in the pancreas of recovering STZ-induced DR in response to GE through immunohistochemistry using various pancreatic endocrine and β-cell markers”. Since this study had a limited budget, the aim was to investigate NMP factors using real-time PCR in the pancreas and liver. Immunostaining for mature beta cell markers like proinsulin, insulin among other markers was performed in another study (different grant) where the pancreatic islets were comprehensively investigated. A manuscript was generated and submitted from that study.
8. To figure out the source of those new beta cells, they should measure the gene expression levels of Nkx6.1 (mature beta cells marker) after 1 and 8 weeks of GE treatment.
- In a current study (new grant), we are investigating the source of new beta cells using a panel of mature beta cell markers. At the time when the current study was completed, we had a limited budget, but the data generated was interesting. This encouraged us to apply for another bigger grant to carry out a more comprehensive investigation.
9. In Figure 4, they use whole pancreas tissue as samples for gene expression measurement, it’s hard to exclude the other cell type (exocrine cells) interference due to pancreas heterogeneity. They should isolate islet cells for the samples.
- In this study we aimed to only investigate the expression of NMP factors using real-time PCR in the pancreas and liver since the grant was limited. Moreover, even when using whole pancreas, we did find a significant initial increase of all 3 genes (MafA, Ngn3, and Pdx1) in the pancreas after only 1 week of GE treatment and a possible explanation of this initial increase was provided in the manuscript. Specific endocrine pancreatic cellular source as well as other cell types like pancreatic exocrine cells are being investigated in an ongoing study from a different grant.
10. To address their hypothesis, there are many experiments they can do, such as primary islet cell and hepatocytes in vitro cultured with GE then collect the sample for RT-PCR, measurement of those transcription factors (Ngn3, Pdx1, MafA, and Nkx6.1).
- Our established recovering diabetic rat model provides a complete physiological environment to investigate the overall effect of GE on alleviating symptoms of diabetes. Therefore, blood glucose and serum insulin can be collected together with the differential expression of NMP factors over the span of the treatment period providing a clear insight on the source of insulin and also allowing us to continue a more comprehensive future investigation to look for the origin of insulin producing cells in the pancreas and/or the liver.
10. Please showed all the individual dots in the bar graph
I am not sure which bar graphs you are referring to. I am assuming the bar graphs in figure 4 (A, B, and C). Individual values are available but unfortunately the person is on holiday and does not have access to the data. The tables provided show average values with the standard error of the mean (SEM).
A. The average normalized expression (2-ΔΔCt) values and SEM for Ngn3 expression in the pancreas during the 8 weeks of treatment.
Ngn3 Expression in Pancreas (fold-change)
Week Treatment Average Normalize Expression 2-ΔΔCt SEM
1 DR-NS 1.00 0.19
1 DR-GE 6.81 2.12
8 DR-NS 4.25 0.93
8 DR-GE 4.72 1.07
(DR-NS diabetic rats treated with saline, DR-GE diabetic rats treated with garlic extract).
B. The average normalized expression (2-ΔΔCt) values and SEM for Pdx1 expression in the pancreas during the 8 weeks of treatment.
Pdx1 Expression in Pancreas (fold-change)
week Treatment Average Normalize Expression 2-ΔΔCt SEM
1 DR-NS 1.00 0.23
1 DR-GE 1.90 0.28
8 DR-NS 1.34 0.31
8 DR-GE 1.07 0.18
(DR-NS diabetic rats treated with saline, DR-GE diabetic rats treated with garlic extract).
C. The average normalized expression (2-ΔΔCt) values and SEM for MafA expression in the pancreas during the 8 weeks of treatment.
MafA Expression in Pancreas (fold-change)
Week Treatment Average Normalize expression 2-ΔΔCt SEM
1 DR-NS 1.00 0.30
1 DR-GE 53.20 9.02
8 DR-NS 4.86 4.22
8 DR-GE 3.19 0.97
(DR-NS diabetic rats treated with saline, DR-GE diabetic rats treated with garlic extract).
Round 2
Reviewer 1 Report
The authors replied in a satisfactory manner to the main issue raised in the first version.
Author Response
There are no further points to address as the referee is satisfied with our response.
Reviewer 2 Report
Please address those two questions:
4. Please provide the individual body weight (g) in supplements.
11. Please showed all the individual dots in the bar graph.
Author Response
We tried to retrieve the raw data but fortunately couldn't. After analyzing all the data and sending the manuscript to Genes, our lab moved to the new campus (from Khaldiya campus to the new Sabah Al-Salem Kuwait University city campus) and It seems that the raw data was lost during the lab relocation. Moreover, I have replied fully to 9 of the 11 comments except for the raw data that was asked for. Again, please accept our apologies.
- Please provide the individual body weight (g) in Figure 1.
The table provided shows body weight percentages with the standard error of the mean (SEM), values in (g) are available but unfortunately the person is on holiday and does not have access to the data).
- Rats body weight were measured and saved as percentages with SEM.
BODY WEIGHT (GAIN/LOSS %)
Group Pre STZ Post STZ (n=12 ) Week 1 (n= 12) Week 8 (n=4)
DR-NS 100 89 ± 0.879 82 ± 0.743 50 ± 0.087
DR-GE 100 83.19 ± 0.789 87.2 ± 0.398 119.21 ± 0.934
(DR-NS diabetic rats treated with saline, DR-GE diabetic rats treated with garlic extract). -
8. Please showed all the individual dots in the bar graph
I am not sure which bar graphs you are referring to. I am assuming the bar graphs in figure 4 (A, B, and C). Individual values are available but unfortunately the person is on holiday and does not have access to the data. The tables provided show average values with the standard error of the mean (SEM).
A. The average normalized expression (2-ΔΔCt) values and SEM for Ngn3 expression in the pancreas during the 8 weeks of treatment.
Ngn3 Expression in Pancreas (fold-change)Week Treatment Average Normalize Expression 2-ΔΔCt SEM
1 DR-NS 1.00 0.19
1 DR-GE 6.81 2.12
8 DR-NS 4.25 0.93
8 DR-GE 4.72 1.07
(DR-NS diabetic rats treated with saline, DR-GE diabetic rats treated with garlic extract).B. The average normalized expression (2-ΔΔCt) values and SEM for Pdx1 expression in the pancreas during the 8 weeks of treatment.
Pdx1 Expression in Pancreas (fold-change)
week Treatment Average Normalize Expression 2-ΔΔCt SEM
1 DR-NS 1.00 0.23
1 DR-GE 1.90 0.28
8 DR-NS 1.34 0.31
8 DR-GE 1.07 0.18(DR-NS diabetic rats treated with saline, DR-GE diabetic rats treated with garlic extract).
C. The average normalized expression (2-ΔΔCt) values and SEM for MafA expression in the pancreas during the 8 weeks of treatment.
MafA Expression in Pancreas (fold-change)
Week Treatment Average Normalize expression 2-ΔΔCt SEM
1 DR-NS 1.00 0.30
1 DR-GE 53.20 9.02
8 DR-NS 4.86 4.22
8 DR-GE 3.19 0.97
(DR-NS diabetic rats treated with saline, DR-GE diabetic rats treated with garlic extract).